# Crohn’s Disease Susceptibility and Onset Are Strongly Related to Three *NOD2* Gene Haplotypes

**DOI:** 10.3390/jcm10173777

**Published:** 2021-08-24

**Authors:** Marta Kaczmarek-Ryś, Szymon Tytus Hryhorowicz, Emilia Lis, Tomasz Banasiewicz, Jacek Paszkowski, Maciej Borejsza-Wysocki, Jarosław Walkowiak, Wojciech Cichy, Piotr Krokowicz, Elżbieta Czkwianianc, Andrzej Hnatyszyn, Iwona Krela-Kaźmierczak, Agnieszka Dobrowolska, Ryszard Słomski, Andrzej Pławski

**Affiliations:** 1Institute of Human Genetics, Polish Academy of Sciences, 60-479 Poznan, Poland; szymon.hryhorowicz@igcz.poznan.pl (S.T.H.); emilia.lis@igcz.poznan.pl (E.L.); slomski@up.poznan.pl (R.S.); 2Department of General, Endocrinological Surgery and Gastroenterological Oncology, Poznan University of Medical Sciences, 60-355 Poznan, Poland; tbanasie@ump.edu.pl (T.B.); jpaszkow@ump.edu.pl (J.P.); maciejbw@ump.edu.pl (M.B.-W.); 3Department of Pediatric Gastroenterology and Metabolic Diseases, Poznan University of Medical Sciences, 60-572 Poznan, Poland; jarwalk@ump.edu.pl (J.W.); wcichy@ump.edu.pl (W.C.); 4Department of General and Colorectal Surgery, Poznan University of Medical Sciences, 61-285 Poznan, Poland; piotr.krokowicz@ump.edu.pl; 5Department of Gastroenterology, Allergology and Pediatrics, Polish Mother’s Memorial Hospital–Research Institute, 93-338 Lodz, Poland; sek41@iczmp.edu.pl; 6Independent Public Health Care Centre in Nowa Sol, Multispecialty Hospital, 67-100 Nowa Sol, Poland; andrzej.hnatyszyn@gmail.com; 7Department of Gastroenterology, Dietetics and Internal Diseases, Poznan University of Medical Sciences, 60-355 Poznan, Poland; krela@op.pl (I.K.-K.); agdob@ump.edu.pl (A.D.)

**Keywords:** Crohn’s disease, ulcerative colitis, disease susceptibility, disease onset, *NOD2* gene, DNA sequence variants, allele distribution, haplotypes, population genetics

## Abstract

The genetic background and the determinants influencing the disease form, course, and onset of inflammatory bowel disease (IBD) remain unresolved. We aimed to determine the *NOD2* gene haplotypes and their relationship with IBD occurrence, clinical presentation, and onset, analyzing a cohort of 578 patients with IBD, including children, and 888 controls. Imaging or endoscopy with a histopathological confirmation was used to diagnose IBD. Genotyping was performed to assess the differences in genotypic and allelic frequencies. Linkage disequilibrium was analyzed, and associations between haplotypes and clinical data were evaluated. We emphasized the prevalence of risk alleles in all analyzed loci in patients with Crohn disease (CD). Interestingly, c.2722G>C and c.3019_3020insC alleles were also overrepresented in ulcerative colitis (UC). T-C-G-C-insC, T-C-G-T-insC, and T-T-G-T-wt haplotypes were correlated with the late-onset form of CD (OR = 23.01, 5.09, and 17.71, respectively), while T-T-G-T-wt and C-C-G-T-wt were prevalent only in CD children (OR = 29.36, and 12.93, respectively; *p*-value = 0.001). In conclusion, the presence of c.3019_3020insC along with c.802C>T occurred as the most fundamental contributing diplotype in late-onset CD form, while in CD children, the mutual allele in all predisposing haplotypes was the c.2798 + 158T. Identifying the unique, high-impact haplotypes supports further studies of the *NOD2* gene, including haplotypic backgrounds.

## 1. Introduction

The incidence of inflammatory bowel disease (IBD) in Western European populations has been estimated at 100–300 per 100,000 and. most alarmingly, it is still increasing [1]. Despite extensive studies over the past decade [2,3,4,5,6,7], the etiology and pathogenesis of IBD are still not fully understood. Furthermore, the factors influencing the early onset of symptoms of the disease have also not yet been determined. Still, it is now widely accepted that intestinal microbiota, environmental (e.g., diet, smoking, and physiological stress), immunological factors, and genetic susceptibility play crucial roles [6].

Hugot et al., in 1996 [8], identified the 16q12 region of the human genome as the first disease-associated locus (IBD1). The discovery of the *NOD2* gene in this locus led to the subsequent research and identification of IBD-related vulnerability loci. To date, genome-wide association studies (GWASs) and subsequent replication studies have provided further insights on IBD’s pathogenesis by determining over 200 genetic risk loci and over 30 non-conservative mutations in the *NOD2* gene [6,9,10,11,12,13,14]. However, the role of novel IBD-associated loci, of which around 30 are shared between Crohn’s disease (CD) and ulcerative colitis (UC), is still less known than IBD1, within which DNA sequence variations remain the most influential genetic disease-triggering factors.

The *NOD2* gene (dbGene 64127) encodes the nucleotide-binding oligomerization domain-containing protein *2* (NOD2)*,* also known as caspase recruitment domain-containing protein 15 (CARD15). Being responsible for bacterial pathogen recognition and inflammasome initiation by binding to muramyl dipeptide (MDP), the NOD2 protein plays a vital role in the immune system. Changes in the *NOD2* gene sequence affecting the function of the protein product and disturbing the balance of the inflammatory response, as a result, were especially reported in CD patients [6,15]. The three main SNPs (single nucleotide polymorphisms) within the *NOD2* gene are most often described as being strongly associated with a higher incidence of CD, but not UC: SNP8 located in exon 4 (c.2104C>T, p.Arg702Trp), SNP12 in exon 8 (c.2722G>C, p.Gly908Arg), and SNP13 in exon 11 (c.3019_3020insC, p.Leu1007fs) [16]. Not infrequently studied and described in the literature is a common polymorphism in exon 4 of the *NOD2* gene (SNP5), where cytosine is replaced with thymine in c.802C>T (p.Pro268Ser). The prevalence of the p.268Ser allele reaches 49.5% in CD patients compared to 18.6% in the global population (according to the gnomAD database), and determines the higher probability of parenteral symptoms and earlier onset of disease in homozygotes [17]. Some reports indicate a strong link between the c.2798 + 158C>T (*IVS8^+158^*) variant, localized in intron 8, and CD development [18,19]. Initially, the *IVS8^+158^* variant was identified as part of a haplotype in which none of the four above-described mutations occurs [18]. Genetic studies carried out on the Ashkenazi Jewish (AJ) population showed a higher frequency of *NOD2* gene mutations in patients from multiplex families with CD, and a clear link between the incidence of CD and the occurrence of the *IVS8^+158^* variant [20]. However, an additional genetic factor within this haplotype, predisposing to the development of the disease, was proposed [18]. These suggestions can be explained by the fact that CD is particularly frequent in the AJ population, characterized by severe bottlenecks and the cultural rule of endogamy, causing a higher incidence of certain genetically determined diseases than other ethnic groups. Some of the described loci associated with Crohn’s disease in other populations are also related to CD in AJs [21]; nevertheless, the increased prevalence of Crohn’s disease in the Jewish population is unexplained, suggesting potentially rare, AJ-specific genetic variants. In their latest report, Rivas et al. indicated that it is unlikely that the incidence of multigenic diseases will change significantly due to the bottleneck itself. The observed difference in the prevalence of Crohn’s disease combined with the systematic enrichment of risk-increasing alleles is unlikely to have occurred by chance, and suggests an unintended selection of alleles in the AJ population. These authors suggest that a subset of Crohn’s risk alleles may contribute to a typical biological process (e.g., a specific immunological response) or phenotype that has been positively selected in AJ. They admitted that the recent population bottleneck could reveal alleles with a significant increase in frequency, which may consequently contribute to interpopulation differences in genetic susceptibility factors. While the *NOD2* gene variants are significantly associated with the genetic risk of CD, other genes with causal alleles that have not passed through the bottleneck are neglected [22].

The fact that genetic influences and inflows remain small or insignificant within the AJ population is undisputed. Still, we do not currently know how many admixtures of Jewish origin exist in European communities, but must bear in mind the history of Jews settling in Europe before the Second World War, particularly in the Polish population. 

Based on this hypothesis and the newest studies by Horovitz et al., showing recessive inheritance of rare *NOD2* variants in 7–10% of CD cases and indicating *NOD2* as a Mendelian disease gene for early-onset CD [23], we aimed to estimate the frequency of c.802C>T, c.2104C>T, c.2722G>C, c.2798 + 158C>T, and c.3019_3020insC variants in a cohort of Polish IBD patients, including children. Thus, despite several studies considering *NOD2* variants, we are the first to determine the distribution of *NOD2* gene haplotypes in Polish IBD patients as well as those haplotypes’ relationships with the disease occurrence, including its subtype and onset.

## 2. Materials and Methods

### 2.1. Study Group 

The research group comprises IBD patients (adults and children) diagnosed with Crohn’s disease and ulcerative colitis. Adults were patients aged over 16, and patients below 16 were classified as pediatric cases (according to the Montreal criteria) [24]. Patients were subjected, diagnosed, and managed in hospitals and clinics all over Poland, and prospectively included in the study. The inclusion criteria were as follows: diagnosis of IBD based on cross-sectional imaging or endoscopy, with histopathological confirmation or both disease duration over one year and lack of any other autoimmune condition (e.g., rheumatoid arthritis, chronic renal failure). The indeterminate IBD cases were excluded from this research. Healthy unrelated adult individuals attending paternity testing and randomly selected from the Polish population (courtesy of the Laboratory of Molecular Genetics, Poznan, Poland) constituted the control group. The number of study groups differed depending on the analyzed polymorphism. The IBD group consisted of 348–578 patients. The control group comprised 231–888 individuals.

### 2.2. Genotyping

Genomic DNA was extracted from peripheral blood leukocytes, following the standard phenol-chloroform procedure, and stored in AE buffer (0.5M Tris-HCl with 0.1M EDTA). We performed pyrosequencing and/or high-resolution melting analysis (HRMA) followed by Sanger sequencing to conduct the linkage study. Primers were designed using PyroMark Assay Design Software 2.0 (Biotage, Uppsala, Sweden) and Primer 3 plus software (Free Software Foundation, Inc., Boston, MA, USA) [25]. Templates for pyrosequencing were amplified by PCR as follows: a 5 min initial denaturation at 94 °C, then 50 cycles of 94 °C for 30 s, annealing for 30 s, followed by a 5 min final elongation at 72 °C. According to the manufacturer’s recommendations, pyrosequencing reactions were performed on the PSQ96 system using PyroMark Gold Q96 reagents (Qiagen, Hilden, Germany). For HRM, polymerase chain reactions were carried out using a commercially available Type-it HRM kit (Qiagen, Hilden, Germany). The PCR program started with initial denaturation (5 min at 95 °C) followed by 40 cycles of denaturation (10 s at 95 °C), annealing (30 s at 55–64 °C), and elongation (10 s at 72 °C) with 10 min extension incubation at 72 °C. The amplified samples were melted between 70 and 90 °C, raising the temperature by 0.1 °C at each step. For all HRMA data evaluations, we used the Rotor-Gene Q Series Software (Qiagen, Hilden, Germany). Genotypes were evaluated in comparison to control samples determined by Sanger sequencing. Samples with an altered melting profile were confirmed by sequencing samples chosen randomly (Appendix A).

For *NOD2* association analysis, we genotyped all IBD patients and control samples for c.802C>T, c.2104C>T, c.2722G>C, c.2798 + 158C>T, and c.3019_3020insC. Primers for pyrosequencing and HRM used for screening *NOD2* genes are listed in the Appendix A.

### 2.3. Statistical Analyses

We applied the χ2 or Fisher’s exact test to assess differences in genotypic and allelic frequencies between the studied groups. The odds ratios (ORs) with 95% confidence intervals (CIs) were computed using calculators on the websites http://ihg.gsf.de/ihg/snps.html (accessed on 2 November 2020) and https://www.medcalc.org/calc/ (accessed on 8 December 2020). Counted *p*-values below 0.05 were considered statistically significant.

### 2.4. Linkage Disequilibrium Analysis

The linkage disequilibrium analysis for the polymorphisms under study and the associations between the haplotypes and the clinical data were conducted using Haploview v.4.2 software (Broad Institute of MIT and Harvard, Cambridge, MA, USA) [26].

## 3. Results

We assembled a large group of Polish IBD patients, including adults and children, from the Polish population. The following loci localized in the *NOD2* gene were genotyped: c.802C>T (in 556 patients and 598 controls), c.2104C>T (in 575 patients and 539 controls), c.2722G>C (in 578 patients and 715 controls), c.2798 + 158C>T (in 348 patients and 231 controls), and c.3019_3020insC (in 573 patients and 888 controls) (Table 1, Table 2, Table 3, Table 4 and Table 5). 

In the case of Crohn’s disease patients, analyzing c.802C>T, c.2104C>T, and c.2722G>C variants, the study group was 217 adults and 85 children, and the subjects with c.3019_3020insC mutations were 215 adults and 84 children, and 97 adults and 78 children regarding c.2798 + 158C>T polymorphism. In the case of patients with ulcerative colitis, we assessed the c.802C>T variant for 189 adults and 65 children, and c.2104C>T for 210 adults and 65 pediatric patients. Concerning c.2722G>C mutation, 211 adults and 65 children were analyzed; for c.3019_3020insC, the number of patients was 210 adults and 64 pediatric patients, and for c.2798 + 158C>T polymorphism, the number was 111 adults and 62 children. Analyzing controls, the c.802C>T variant was studied in 598 individuals, the c.2104C>T variant in 539 individuals, the c.2722G>C variant in 725 individuals, the c.3019_3020insC variant in 888 individuals, and the c.2798 + 158C>T variant in 231 individuals.

### 3.1. The NOD2 Gene Allele and Genotype Distribution

All of the examined SNPs showed statistically significant differences in allele frequency when comparing IBD patients with controls. We noted the prevalence of risk alleles in patients: the c.802T allele was present in 35.9% of IBD patients and 30.1% of control individuals (OR = 1.30, CI = (1.09–1.55), *p*-value = 0.003), the c.2104T allele was observed in 4.4% of patients and 2.0% of controls (OR = 2.24, CI = (1.16–4.35), *p*-value = 0.014), the c.2722C allele was over three times more frequent in the IBD group (3.7%) than in the population (1.1%) (OR = 3.65, CI = (2.01–6.59), *p*-value < 0.001), the c.2798 + 158T allele occurred in 23% of patients compared to 17.1% of controls (OR = 1.45, CI = (1.07–1.95), *p*-value = 0.015), and c.3019_3020insC was present with three times higher frequency in patients (14.7%) than in controls (4.1%) (OR = 3.01, CI = (3.01–5.33), *p*-value < 0.001) (Table 1, Table 2, Table 3, Table 4 and Table 5).

We analyzed patients with CD and UC independently, comparing them with population controls regarding differences in disease form. In the next step, we set CD and UC against one another. Moreover, we divided the CD and UC groups into adults and pediatric patients, and within these groups, we also considered gender.

In patients with CD, all risk alleles of analyzed loci were much more prevalent than in the general population or in UC patients, and all observations were statistically significant. 

The c.802T allele was present in 44.2% of all CD patients (OR = 1.84, CI = (1.50–2.26), *p*-value < 0.001) and only 26% of the UC group, making the difference between UC patients and the general population insignificant (OR = 0.83, CI = (0.66–1.04), *p*-value = 0.106). Comparing CD with UC patients, the difference was even more visible (OR = 2.26, CI = (1.75–2.91), *p*-value < 0.001). 

Homozygous carriers of the c.802T allele occurred twice as frequently in IBD patients as in controls (OR = 2.00, CI = (1.36–2.93), *p*-value < 0.001). In addition, being homozygous increased CD risk more than 3.5 times in comparison to the analyzed population (OR = 3.70, CI = (2.43–5.63), *p*-value < 0.001).

We did not observe differences between subgroups regarding adult patients and children independently in CD patients. However, looking at UC patients, we noticed relevant differences between pediatric and adult patients: in children, the c.802T allele was more frequent than in adults (OR = 1.69, CI = (1.092–2.604), *p*-value = 0.018, data not shown in tables). Considering gender, we did not observe differences in the adult patient groups. However, the c.802T allele seems to be significantly (*p*-value = 0.026) more prevalent in UC boys in comparison to girls (41.9% vs. 23.2%, OR = 2.39, CI = (1.100–5.168)), as well as in CD children (47.2% vs. 41.5%, respectively), where it did not meet the assumed level of statistical significance (*p*-value = 0.411) (results not included in tables).

In the case of the c.2104T allele, we also observed its higher impact in CD patients (OR = 3.61, CI = (1.84–7.09), *p*-value < 0.001), and no significant effect on UC susceptibility (*p*-value = 0.670). We did not observe any homozygotes in this locus. The differences between adults and children and between genders were noticeable in the UC group only (3.1% vs. 1.2%), but statistically insignificant.

In the rs2066845 locus we observed allele c.2722C in 3.7% of IBD patients and in only 1.1% of controls (OR = 3.65, CI = (2.01–6.59), *p*-value < 0.001). Analysis of the c.2722C allele carriers in the patient group showed its higher relationship with CD risk (OR = 4.24, CI = (2.23–8.07), *p*-value < 0.001), but also a significant association with UC development (OR = 3.00, CI = (1.49–6.05), *p*-value = 0.001). Looking more closely at pediatric patients, we found that in CD children, the c.2722C allele occurred with a higher frequency than in adults (5.9% vs. 3.7%); in comparison to the population, it was almost six times more prevalent (OR = 5.89, CI = (2.60–13.34), *p*-value < 0.001). 

In UC children, in contrast, the c.2722C allele was underrepresented in comparison to the general population and UC adults (0.8% vs. 1.1% and 3.8%, respectively); however, these discrepancies did not meet the assumed level of statistical significance. We did not notice significant differences in allele distribution between genders either. Being homozygous in the rs2066845 locus raised CD risk 10-fold (OR = 10.38, CI = (0.42–255.72)), but this result was not significant since we detected only one homozygous female CD patient in the tested group. 

The c.2798 + 158T variant was 1.5 times more frequent comparing IBD patients with controls (OR = 1.45, CI = (1.07–1.95); *p*-value = 0.015); nevertheless, after considering disease entity, it applied to CD patients only (OR = 2.14, CI = (1.53–2.98), *p*-value < 0.001). Moreover, when juxtaposing CD children with UC patients, the observed difference was clearer (OR = 2.43, CI = (1.68–3.53), *p*-value < 0.001).

Allele c.3019_3020insC frequency analysis revealed three times higher occurrence in the IBD group than in the controls (OR = 3.01, CI = (3.01–5.33), *p*-value < 0.001), which was highlighted after classification versus disease entities: in Crohn’s patients, we observed six times higher incidence of this allele (OR = 6.04, CI = (4.44–8.21), *p*-value < 0.001). Moreover, after dividing the CD group into adult and pediatric patients, allele c.3019_3020insC was seven (OR = 6.89, CI = (4.98–9.53), *p*-value < 0001) and four times (OR = 4.08, CI = (2.51–6.63), *p*-value < 0.001) more frequent than in the control group, respectively. Analyzing the genotypes in the rs5743293 locus, we observed that the homozygous c.3019_3020insC genotype occurred in 39 IBD patients (6.8%) and not once in the control group (OR = 64.16, CI = 8.38–2228.1, *p*-value < 0.001). Regarding disease entity, in adult Crohn’s patients, the homozygous insC/insC genotype was present in 11.6% of subjects (OR = 291.86, CI = 17.67–4821.24, *p*-value < 0.001), and was significantly more frequent than in adult patients with UC, of whom only 2.6% were homozygous (OR = 5.19, CI = 2.24–12.00, *p*-value < 0.001). Nevertheless, in UC adults, homozygous insertion genotype still presented relevant overrepresentation compared to controls (OR = 57.46, CI = 3.22–1024.52, *p*-value < 0.001).

### 3.2. Linkage Disequilibrium (LD) Analysis

Linkage disequilibrium between all pairs of common polymorphic sites was evaluated, with D′ being determined using Haploview v.4.2 (Broad Institute of MIT and Harvard, Cambridge, MA, USA) [26]. A strong linkage disequilibrium (D′ = 1) was observed between two pairs of markers: c.802C>T (p.Pro268Ser) with c.2722G>C (p.Gly908Arg) (pediatric and adult IBD patients); and c.802C>T (p.Pro268Ser) with c.2104C>T (p.Arg702Trp) (in pediatric patients only). In the case of another marker pair observed in both studied groups—c.802C>T (p.Pro268Ser) and c.2798 + 158C>T (IVS8^+158^)—we noticed LD, at the level of D = 0.79 and D = 0.87, respectively. Interestingly, variant c.802C>T (p.Pro268Ser) showed strong LD with all markers (mean D = 0.90, range 0.79–1.00), including c.3019_3020insC (p.Leu1007Pro_fs) (D = 0.83 in pediatric IBD patients) (Figure 1).

In both IBD groups, the most frequent allele combinations were as follows: C-C-G-C-wt (A haplotype according to Tukel et al. [20], 62.3% adults, 54.7% children), T-C-G-C-wt (H haplotype according to Tukel et al. [20], 8.6% adults, 10.9% children), T-C-G-T-wt (G haplotype according to Tukel et al. [20], 7.8% adults, 10.1% children), and T-C-G-T-insC (E haplotype according to Tukel et al. [20], 7.5% adults and 9.5% children).

For haplotype gathering of wild-type alleles only, C-C-G-C-wt was the most common in patients and in the control group. Surprisingly, however, it was present in patients with even higher frequency than in the general population (OR = 3.40, CI = (2.43–4.78) and OR = 1.77, CI = (1.27–48), respectively), and it was statistically significant (*p*-value < 0.001). We found five allele sets in the assessed control group that did not appear in IBD patients, and three rare haplotypes present only in the IBD group.

In the next step, we analyzed haplotypes’ association with disease entity, age of first symptoms, and sex (Table 6).

We noticed that the haplotypes: T-C-G-T-insC (E haplotype according to Tukel et al. [20]), T-C-G-C-insC (unreported earlier), and T-T-G-T-wt (B haplotype according to Tukel et al. [20]) correlated with the late-onset form of CD (ORs were 5.09, 23.01, and 17.71, respectively) and observations were statistically relevant (*p*-values were < 0.001, 0.003, and 0.007, respectively). Interestingly, three unreported earlier rare allele combinations were present exclusively in IBD patients: T-C-G-C-insC was characteristic only of adult CD and pediatric UC patients. In contrast, T-T-G-T-wt was more prevalent in CD children than in adults, with frequency 30 times higher than in the general population (OR = 29.36, CI = (3.69–233.68), *p*-value = 0.001). Furthermore, the haplotype C-C-G-T-wt was present exclusively in CD pediatric cases, making it predisposing to early-onset CD (OR = 12.93, CI = (2.72–61.58), *p*-value = 0.001).

The haplotype T-C-G-T-wt was approximately twice as frequent in both IBD groups as in the population. It met the statistical significance level in UC patients (adults and children) and CD children, while it was borderline significant in CD adults. The haplotypes C-C-C-C-wt, C-C-C-T-wt, C-C-C-C-insC, T-C-C-T-insC, and T-C-C-T-wt (all unreported by Tukel et al. [20]) were absent in IBD patients. However, they were present in the tested population group, and C-C-C-C-wt was the second most frequent (OR calculations were possible using Deek’s correction [27]).

## 4. Discussion

Allelic variants of the *NOD2* gene (c.802C>T, c.2104C>T, c.2722G>C, c.2798 + 158C>T, c.3019_3020insC) are generally overrepresented in the Polish population in comparison with other ethnic groups (Table 7).

Still, in CD patients, the risk alleles are prevalent. Moreover, their frequencies are different comparing patients with CD and UC. This observation led us to conclude that the *NOD2* gene and its sequence variations remain among the most critical genetic backgrounds of IBD, contributing particularly to susceptibility to Crohn’s disease.

Two decades of research and numerous scientific reports have not yet unraveled the molecular basis of IBD. They have shown that CD and UC are distinct diseases conditioned by multiple genetic factors [21,22,23,24]; nevertheless, the *NOD2* gene and its sequence variations remain the most influential genetic disease-triggering factors. To date, ~2404 variants of the *NOD2* gene have been described together with specific phenotypes. Earlier studies have shown that, among the European and North American populations, the most frequent changes in the *NOD2* gene associated with CD are 1007fs, R702W, and G908R [19,28], but also the c.802C>T variant [16,29]. In general, mutations in the *NOD2* gene are indicated as being CD-related in several Caucasian populations [30]. On the other hand, association studies among Indian IBD patients have shown a weak relationship of the *NOD2* gene mutations with UC, but not with CD [31]. In turn, the study of *NOD2* gene variants in patients with UC in the Portuguese population did not correlate them with increased risk of the disease; however, a tendency for a more aggressive course of the disease was observed among carriers of rare *NOD2* variants [32].

In this research, we highlighted the prevalence of risk alleles in all analyzed loci in CD patients; however, the c.2722G>C and c.3019_3020insC alleles were unexpectedly overrepresented in UC patients as well. Moreover, gathering of pediatric IBD cases and linkage disequilibrium analysis in this particular group of patients enabled us to observe possible relationships of specific haplotypes with early disease onset.

It is widely known that analyzing haplotypes enables the observation of correlations that may not be apparent for single markers. However, we did not find many papers describing *NOD2* haplotypes in the context of IBD susceptibility and course. Many of the articles describe variants in the *NOD2* gene considered individually. In the newest study performed on a large cohort of CD patients, Horowitz et al. reported that individuals carrying any one of the main three *NOD2* risk alleles (p.R702W, p.G908R, or p.L1007fs) have up to 4-fold increased risk for developing CD, while carriers of two or more of the same *NOD2* variants have 15–40-fold increased risk. Moreover, these authors highlighted a subset of IBD cases with the recessive inheritance of *NOD2* alleles and a substantially higher risk of early CD onset; their analyses unequivocally showed more significant effects for *NOD2* homozygotes and compound heterozygotes than carriers of single *NOD2* genetic variants only, and indicated that the genetic contribution of *NOD2* alleles, in a subset of Crohn’s disease patients, suggests a recessive disease model [23]. The present results entirely support the conclusions of Horowitz et al.

In 2004, Tukel et al. described *NOD2* haplotypes detected in AJ and Sephardi/Oriental Jewish (SOJ) populations; they also presented possible evolution of haplotypes, and confirmed their theory by analyzing flanking STR markers [20]. In our study, we detected all allele combinations described in AJs and SOJs; nevertheless, we also identified new haplotypes, indicating possible recombination events in this region (Table 6).

To date, the highest recorded incidence of IBD has been reported in non-Hispanic whites (inhabiting Central and Western Europe and North America)—three times higher than in other ethnic groups [33]. In this population, a strong association between mutant *NOD2* and CD risk was reported for R702W, G908R, and 1007insC when they were considered separately (*p*-values: < 0.001, 0.002, and < 0.001, respectively) [34]. G908R and L1007fs were associated with Crohn’s disease susceptibility in the Dutch population, and carrying of at least one of these mutations was associated with more severe and penetrating disease [35]. In Hungarian adult Crohn’s patients, the 1007finsC and the 1007finsC and G908R in the pediatric cases were significantly associated with increased disease risk [36]. In Greek CD patients, the 1007finsC mutation was significantly more frequent in childhood-onset than in adult-onset form [37]. Other research carried out in Greece pointed out the association of R702W, G908R, and 1007insC with ileitis or ileocolitis in the clinical picture of CD [38,39]. Similarly, in an Italian multicenter study, Ferraris et al. demonstrated that *NOD2* polymorphisms were associated with susceptibility to early-onset CD, and with ileal involvement. For the first time, they also reported an association with severe, early-onset UC [40]. Regarding the group of UC children presented in this study, we obtained similar results, with a relevantly higher frequency of the c.3019_3020insC allele (OR = 2.64) than in the general population, and this was visible in UC adults as well (OR = 1.92). However, regarding haplotypes, the presence of insertion mutation alone was not sufficient to increase IBD susceptibility. The definite positive effect was only observed when c.3019_3020insC was accompanied by the c.802C>T variant; nevertheless, in UC children, it did not reach the statistical significance level. In UC adult and pediatric patients, the T-C-G-T-wt haplotype seemed to be predisposing in our study cohort (OR = 2.00 and 2.61, respectively). Since the haplotype including only wt alleles (C-C-G-C-wt) was in the majority of our UC group, this suggests that other variants in the *NOD2* gene may play a substantial role in these patients.

In CD patients, the c.3019_3020insC allele present amidst haplotypes containing wt alleles in the remaining loci seemed not to play a crucial role in disease susceptibility. In contrast, the presence of c.3019_3020insC along with c.802C>T occurred as the most fundamental contributing diplotype in late-onset CD. Furthermore, it appeared to be more complex in CD children in whom the mutual allele in all predisposing haplotypes was the c.2798 + 158T allele. Based on these observations, we are convinced that further studies of *NOD2* gene variants should include haplotypic backgrounds.

Crohn’s disease occurs with the highest frequency in AJs of Central European origin, which is 2–4-fold higher than in non-Jewish ethnic groups [41,42]. AJs are individuals of Jewish ancestry with a recent origin in Central and Eastern Europe. Tukel et al. established that minor alleles of the *NOD2* gene in AJ CD patients from Central Europe are twice as frequent as in AJ patients from Eastern Europe—particularly G908R and 1007fs. Surprisingly, in SOJ patients, *NOD2* mutations were also overrepresented. Moreover, this study’s haplotype analysis revealed that the p.702W allele was associated with the p.268P and p.268S alleles [20]; this was the opposite of the findings of Sugimura et al. (2003), who indicated 702W, 906R, and 1007fs with 268S variants exclusively [42]. The study of Lesage et al. (2002) indicated that 49% of the patients with familial CD had one (32%) or two (17%) mutant *NOD2* alleles; in their research, detected mutations in the majority were localized in the distal part of the gene, and the three common mutations (R702W, G908R, and 1007fs) accounted for 81% of identified alterations [16]. In a different study, 32% of AJ families and 30% of SOJ families with CD did not carry an *NOD2* mutation, indicating the heterogeneity of the predisposing genes causing CD among Jewish patients [20]. Considering the examples mentioned above, we can state that the higher incidence of several diseases—including IBD—in AJs is likely due to genetic drift following a bottleneck; the AJ population is much larger and experienced a more severe bottleneck than other founder populations [43].

We also reflect on the probability of the *NOD2* gene mutations of similar origin in Polish and Jewish populations. We believe that the *NOD2* variants determining the occurrence of Crohn’s disease came from common ancestors, resulting from mutual history. In the 13th century, Ashkenazi communities emerged in Poland and multiplied until the 20th century, reaching millions in size and a wide geographic spread across Europe [44,45]. Assessing genetic distance, Atzmon et al. showed that the AJs are more closely related to some host Europeans than to the ancestral Levantines [44]. Hue et al. suggested a model of at least two events of European admixture: The first of them slightly pre-dated a late medieval founder event, was probably of Southern European origin, and was estimated to be 25 ± 50 generations ago. The inferred subsequent admixture was hypothesized to have appeared approximately 30 generations ago, and most likely occurred in Eastern Europe. However, multiple lines of evidence suggest that it represents an average over two or more events, pre-dating and post-dating the founder event experienced by AJs in the late Middle Ages [45]. Before World War II, approximately three million Jews lived in Poland. After 1945, the most significant number of Jews was recorded in July 1946, and amounted to ~220,000. In the structure of this decimated population, men predominated, causing an increase in mixed marriages, which was one ground for the postwar conversion to Christianity. Moreover, the postwar anti-Semitic moods and the policies of the communist authorities—and in many cases also the adoption of Polish names and surnames—were among the reasons for the decision to change personal details [46,47]. This historical background may explain the higher frequency of minor *NOD2* gene alleles in the Polish population and in Polish patients with CD.

The conclusion from these scientific reports is quite apparent, and widely known: populations of different ethnic origins or living in specific regions show diversity in the distribution of genotypes of a given variant. Most racial and ethnic research on IBD includes analysis of Caucasian populations. Unfortunately, these are usually small groups of subjects characterized by exceptionally high social and economic heterogeneity.

Ng et al. [1] demonstrated that since 1990, incidence rates have changed in Western countries, showing a stable or decreasing incidence. Still, the burden of disease remains high in most European countries, North America, and Oceania. On the other hand, the countries of Africa, the Middle East, Asia, and South America, whose societies are becoming more western and urbanized, reflect the progress in inflammatory bowel diseases in the Western world since the 1900s, which indicates the significant role of environmental externalities in the pathogenesis of the disease.

New locus mapping in GWAS studies leads to the identification of an increasing number of polymorphisms and haplotypes between different populations, which underlines the role of genetic variability in analyzing the molecular background of multigenic diseases. The phenomenon of higher prevalence of minor *NOD2* gene alleles in the Polish population most likely results from historical conditions. The identification of unique, high-impact haplotypes supports population genetics as being fundamental to unravelling IBD’s molecular background. Our results indicate more significant effects of homozygous and compound heterozygous *NOD2* mutations than single *NOD2* genetic variants in conditioning Crohn’s disease. Moreover, extended *NOD2* haplotype analysis suggests the existence of additional genetic factors remaining in linkage disequilibrium, which may be related to IBD susceptibility and onset in a population-specific manner. *NOD2* gene sequence variants and haplotypes play a crucial role, and should not be underestimated in IBD diagnosis.

## Figures and Tables

**Figure 1 jcm-10-03777-f001:**
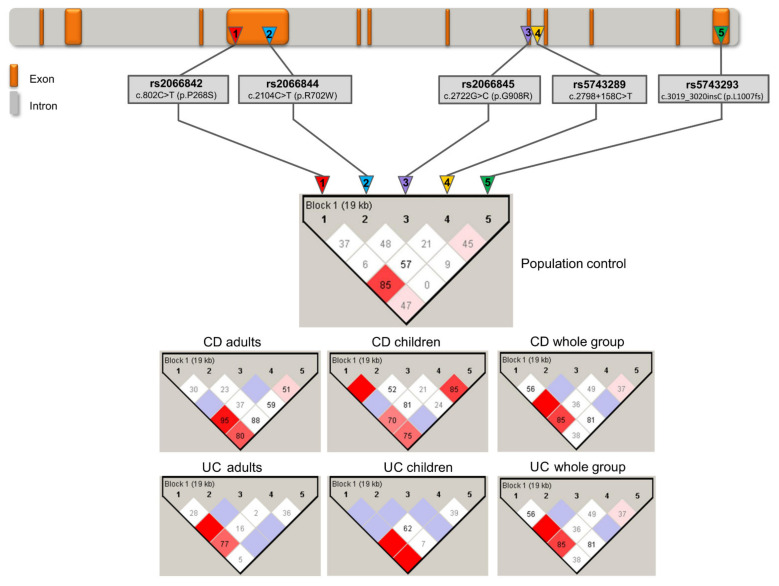
Haplotype analysis of the *NOD2* gene in IBD patients and controls. The top of the figure shows the *NOD2* gene scheme. Analyzed polymorphisms are marked. Tested SNPs were assigned with a number (1–5), which corresponds to the haplotype diagrams presented at the bottom of the figure. Haplotype diagrams concerning all tested groups were prepared with Haploview v.4.2, and show linkage disequilibrium values among SNPs located in the *NOD2* gene. The absolute value of Lewontin’s measure of linkage disequilibrium, D′, was calculated and graphically displayed for all pairs of markers. Numbers in squares represent the D′ statistic × 100 for each pairwise comparison linked by the box if D′ = 1 boxes do not have a shown value. The intensity of the filled-in boxes corresponds with the degree of linkage disequilibrium.

**Table 1 jcm-10-03777-t001:** Allele and genotype distribution for the *NOD2* c.802C>T (p.P268S, rs2066842) variant.

c.802C>T (p.P268S) rs2066842		Genotypes *n* (%)			Alleles *n* (%)	
Group	*n*	TT	CT	CC	T	C
IBD whole group *	556	89 (16.0%)	221 (39.8%)	246 (44.2%)	399 (35.9%)	713 (64.1%)
UC adults	189	13 (6.9%)	62 (32.8%)	114 (60.3%)	88 (23.3%)	290 (76.7%)
UC adult women	78	4 (5.1%)	27 (34.6%)	47 (60.3%)	35 (22.4%)	121 (77.6%)
UC adult men	111	9 (8.1%)	35 (31.5%)	67 (60.4%)	53 (23.9%)	169 (76.1%)
UC children	65	5 (7.7%)	34 (52.3%)	26 (40%)	44 (33.8%)	86 (66.2%)
UC girls	28	1 (3.6%)	11 (39.3%)	16 (57.1%)	13 (23.2%)	43 (76.8%)
UC boys	37	4 (10.8%)	23 (62.1%)	10 (27.1%)	31 (41.9%)	43 (58.1%)
UC whole group	254	18 (7.1%)	96 (37.8%)	140 (55.1%)	132 (26.0%)	376 (74.0%)
CD adults *	217	50 (23.1%)	91 (41.9%)	76 (35%)	191 (44.1%)	243 (55.9%)
CD adult women	117	24 (20.5%)	54 (46.2%)	39 (33.3%)	102 (43.6%)	132 (56.4%)
CD adult men *	100	26 (26%)	37 (37%)	37 (37%)	89 (44.5%)	111 (55.5%)
CD children	85	21 (24.7%)	34 (40%)	30 (35.3%)	76 (43.7%)	98 (56.3%)
CD girls	41	8 (19.5%)	18 (43.9%)	15 (36.6%)	34 (41.5%)	48 (58.5%)
CD boys	44	13 (29.5%)	16 (36.4%)	15 (34.1%)	42 (47.2%)	47 (52.8%)
CD whole group *	302	71 (23.5%)	125 (41.4%)	106 (35.1%)	267 (44.2%)	337 (55.8%)
Controls	598	52 (8.8%)	250 (42.5%)	287 (48.7%)	354 (30.1%)	824 (69.9%)
		(TT) vs. (CC) OR 95% CI *p*-value	(TC) vs. (CC) OR 95% CI *p*-value	(TT) + (TC) vs. (CC) OR 95% CI *p*-value	(T) vs. (C) OR 95% CI *p*-value	
IBD whole group vs. controls		**OR = 2.00** **CI = (1.36–2.93)** ***p* < 0.001**	OR = 1.03 CI = (0.80–1.32) *p* = 0.808	OR = 1.20 CI = (0.95–1.51) *p* = 0.129	**OR = 1.30** **CI = (1.09–1.55)** ***p* = 0.003**	
UC adults vs. controls		OR = 0.63 CI = (0.33–1.20) *p* = 0.157	**OR = 0.62** **CI = (0.44–0.89)** ***p* = 0.009**	**OR = 0.63** **CI = (0.45–0.87)** ***p* = 0.006**	**OR = 0.71** **CI = (0.54–0.92)** ***p* = 0.011**	
UC children vs. controls		OR = 1.06 CI = (0.39–2.89) *p* = 0.907	OR = 1.50 CI = (0.88–2.57) *p* = 0.137	OR = 1.43 CI = (0.85–2.40) *p* = 0.181	OR = 1.19 CI = (0.81–1.75) *p* = 0.372	
UC adults vs. UC children		OR = 0.57 CI = (0.18–1.71) *p* = 0.311	**OR = 0.47** **CI = (0.26–0.84)** ***p* = 0.009**	**OR = 0.48** **CI = (0.27–0.84)** ***p* = 0.010**	OR = 0.62 CI = (0.40–0.95) *p* = 0.026	
UC women vs. UC men		OR = 0.62 CI = (0.18–2.13) *p* = 0.445	OR = 1.10 CI = (0.59–2.06) *p* = 0.766	OR = 1.00 CI = (0.56–1.82) *p* = 0.989	OR = 0.91 CI = (0.56–1.47) *p* = 0.695	
UC girls vs. UC boys		OR = 0.16 CI = (0.01–1.60) *p* = 0.087	**OR = 0.30** **CI = (0.10–0.87)** ***p* = 0.024**	**OR = 0.28** **CI = (0.10–0.79)** ***p* = 0.014**	OR = 0.42 CI = (0.19–0.90) *p* = 0.025	
UC whole group vs. controls		OR = 1.13 CI = (0.53–2.40) *p* = 0.755	OR = 1.26 CI = (0.84–1.89) *p* = 0.260	OR = 1.24 CI = (0.85–1.81) *p* = 0.274	OR = 1.16 CI = (0.85–1.58) *p* = 0.357	
CD adults vs. controls		**OR = 3.63** **CI = (2.28–5.77)** ***p* < 0.001**	OR = 1.38 CI = (0.97–1.95) *p* = 0.073	**OR = 1.76** **CI = (1.28–2.43)** ***p* < 0.001**	**OR = 1.83** **CI = (1.46–2.29)** ***p* < 0.001**	
CD children vs. controls		**OR = 3.86** **CI = (2.05–7.26)** ***p* < 0.001**	OR = 1.30 CI = (0.77–2.19) *p* = 0.320	**OR = 1.74** **CI = (1.09–2.80)** ***p* = 0.020**	**OR = 1.88** **CI = (1.36–2.61)** ***p* < 0.001**	
CD adults vs. UC children		OR = 0.94 CI = (0.48–1.82) *p* = 0.854	OR = 1.06 CI = (0.59–1.88) *p* = 0.852	OR = 1.01 CI = (0.60–1.71) *p* = 0.965	OR = 0.97 CI = (0.68–1.39) *p* = 0.876	
CD women vs. CD men		OR = 0.87 CI = (0.43–1.79) *p* = 0.715	OR = 1.39 CI = (0.75–2.56) *p* = 0.299	OR = 1.18 CI = (0.67–2.05) *p* = 0.573	OR = 0.96 CI = (0.66–1.41) *p* = 0.849	
CD girls vs. CD boys		OR = 0.61 CI = (0.19–1.91) *p* = 0.400	OR = 1.13 CI = (0.42–3.01) *p* = 0.814	OR = 0.90 CI = (0.37–2.18) *p* = 0.810	OR = 0.77 CI = (0.42–1.42) *p* = 0.411	
CD whole group vs. controls		**OR = 3.70** **CI = (2.43–5.63)** ***p* < 0.001**	OR = 1.35 CI = (0.99–1.85) *p* = 0.055	**OR = 1.76** **CI = (1.32–2.34)** ***p* < 0.001**	**OR = 1.84** **CI = (1.50–2.26)** ***p* < 0.001**	
CD whole group vs. UC whole group		**OR = 5.21** **CI = (2.93–9.26)** ***p* < 0.001**	**OR = 1.72** **CI = (1.19–2.48)** ***p* = 0.004**	**OR = 2.27** **CI = (1.61–3.20)** ***p* < 0.001**	**OR = 2.26** **CI = (1.75–2.91)** ***p* < 0.001**	

Statistically significant results are marked in bold; *: deviation from HWE. IBD, inflammatory bowel disease; UC, ulcerative colitis; CD, Crohn’s disease; OR, odds ratio; CI, confidence interval.

**Table 2 jcm-10-03777-t002:** Allele and genotype distribution for the *NOD2* c.2104C>T (p.R702W, rs2066844) variant.

c.2104C>T (p.R702W) rs2066844		Genotypes *n* (%)			Alleles *n* (%)	
Group	*n*	TT	TC	CC	T	C
IBD whole group	575	n.o.	50 (8.7%)	525 (91.3%)	50 (4.4%)	1100 (95.6%)
UC adults	210	n.o.	5 (2.4%)	204 (97.6%)	5 (1.2%)	413 (98.8%)
UC adult women	98	n.o.	1 (1.0%)	96 (99.0%)	1 (0.5%)	193 (99.5%)
UC adult men	112	n.o.	4 (3.6%)	108 (96.4%)	4 (1.8%)	220 (98.2%)
UC children	65	n.o.	4 (6.2%)	61 (93.8%)	4 (3.1%)	126 (96.9%)
UC girls	28	n.o.	2 (7.2%)	26 (92.8%)	2 (3.6%)	54 (96.4%)
UC boys	37	n.o.	2 (5.4%)	35 (94.6%)	2 (2.7%)	72 (97.3%)
UC whole group	275	n.o.	9 (3.3%)	265 (96.7%)	9 (1.6%)	539 (98.4%)
CD adults	217	n.o.	30 (13.9%)	186 (86.1%)	30 (6.9%)	402 (93.1%)
CD adult women	117	n.o.	16 (13.7%)	101 (86.3%)	16 (6.8%)	218 (93.2%)
CD adult men	100	n.o.	14 (14.1%)	85 (85.9%)	14 (7.1%)	184 (92.9%)
CD children	85	n.o.	11 (12.9%)	74 (87.1%)	11 (6.5%)	159 (93.5%)
CD girls	41	n.o.	3 (7.3%)	38 (92.7%)	3 (3.7%)	79 (96.3%)
CD boys	44	n.o.	8 (18.2%)	36 (81.8%)	8 (9.1%)	80 (90.9%)
CD whole group	302	n.o.	41 (13.6%)	260 (86.4%)	41 (6.8%)	561 (93.2%)
Controls	539	n.o.	11 (4.0%)	266 (96.0%).	11 (2.0%)	543 (98.0%)
		(TT) vs. (CC) OR 95% CI *p*-value	(TC) vs. (CC) OR 95% CI *p*-value	(TT) + (TC) vs. (CC) OR 95% CI *p*-value	(T) vs. (C) OR 95% CI *p*-value	
IBD whole group vs. controls		n.a.	**OR = 2.30** **CI = (1.18–4.50)** ***p* = 0.012**	n.a.	**OR = 2.24** **CI = (1.16–4.35)** ***p* = 0.014**	
UC adults vs. controls		n.a.	OR = 0.59 CI = (0.20–1.73) *p* = 0.334	n.a.	OR = 0.60 CI = (0.21–1.73) *p* = 0.338	
UC children vs. controls		n.a.	OR = 1.59 CI = (0.49–5.15) *p* = 0.439	n.a.	OR = 1.57 CI = (0.49–5.00) *p* = 0.502	
UC adults vs. UC children		n.a.	OR = 0.37 CI = (0.10–1.44) *p* = 0.137	n.a.	OR = 0.38 CI = (0.10–1.44) *p* = 0.263	
UC women vs. UC men		n.a.	OR = 0.28 CI = (0.03–2.56) *p* = 0.231	n.a.	OR = 0.29 CI = (0.03–2.57) *p* = 0.382	
UC girls vs. UC boys		n.a.	OR = 1.35 CI = (0.18–10.19) *p* = 0.773	n.a.	OR = 1.33 CI = (0.18–9.77) *p* = 1.005	
UC whole group vs. controls		n.a.	OR = 0.82 CI = (0.34–2.01) *p* = 0.667	n.a.	OR = 0.82 CI = (0.34–2.01) *p* = 0.670	
CD adults vs. controls		n.a.	**OR = 3.90** **CI = (1.91–7.98)** ***p* < 0.001**	n.a.	**OR = 3.68** **CI = (1.82–7.44)** ***p* < 0.001**	
CD children vs. controls		n.a.	**OR = 3.60** **CI = (1.50–8.62)** ***p* = 0.002**	n.a.	**OR = 3.42** **CI = (1.45–8.02)** ***p* = 0.003**	
CD adults vs. CD children		n.a.	OR = 1.09 CI = (0.52–2.28) *p* = 0.829	n.a.	OR = 1.08 CI = (0.53–2.21) *p* = 0.835	
CD women vs. CD men		n.a.	OR = 0.96 CI = (0.44–2.08) *p* = 0.921	n.a.	OR = 0.97 CI = (0.46–2.03) *p* = 0.924	
CD girls vs. CD boys		n.a.	OR = 0.36 CI = (0.09–1.45) *p* = 0.136	n.a.	OR = 0.38 CI = (0.10–1.48) *p* = 0.150	
CD whole group vs. controls		n.a.	**OR = 3.81** **CI = (1.92–7.58)** ***p* < 0.001**	n.a.	**OR = 3.61** **CI = (1.84–7.09)** ***p* < 0.001**	
CD whole group vs. UC whole group		n.a.	**OR = 4.64** **CI = (2.21–9.75)** ***p* < 0.001**	n.a.	**OR = 4.38** **CI = (2.11–9.09)** ***p* < 0.001**	

Statistically significant results are marked in bold; n.o.: not observed; n.a.: not analyzed (TT genotype was not detected).

**Table 3 jcm-10-03777-t003:** Allele and genotype distribution for the *NOD2* c.2722G>C (p.G908R, rs2066845) variant.

c.2722G>C (p.G908R) rs2066845		Genotypes *n* (%)			Alleles *n* (%)	
Group	*n*	CC	GC	GG	C	G
IBD whole group	578	1 (0.2%)	41 (7.1%)	536 (92.7%)	43 (3.7%)	1113 (96.3%)
UC adults	211	n.o.	16 (7.6%)	195 (92.4%)	16 (3.8%)	406 (96.2%)
UC adult women	99	n.o.	7 (7.1%)	92 (92.9%)	7 (3.5%)	191 (96.5%)
UC adult man	112	n.o.	9 (8.1%)	103 (91.9%)	9 (4%)	215 (96%)
UC children	65	n.o.	1 (1.5%)	64 (98.5%)	1 (0.8%)	129 (99.2%)
UC girls *	28	n.o.	n.o.	28 (100%)	0 (0%)	56 (100%)
UC boys	37	n.o.	1 (2.7%)	36 (97.3%)	1 (1.3%)	73 (98.7%)
UC whole group	276	n.o.	17 (6.2%)	259 (93.8%)	17 (3.1%)	535 (96.9%)
CD adults	217	1 (0.5%)	14 (6.5%)	202 (93%)	16 (3.7%)	418 (96.3%)
CD adult women	117	1 (0.9%)	10 (8.5%)	106 (90.6%)	12 (5.1%)	222 (94.9%)
CD adult men	100	n.o.	4 (4%)	96 (%)	4(2%)	196 (98%)
CD children	85	n.o.	10 (11.8%)	75 (88.2%)	10 (5.9%)	160 (94.1%)
CD girls	41	n.o.	3 (7.3%)	38 (92.7%)	3 (3.7%)	79 (96.3%)
CD boys	44	n.o.	7 (15.9%)	37 (84.1%)	7 (7.9%)	81 (92.1%)
CD whole group	302	1 (0.4%)	24 (7.9%)	277 (91.7%)	26 (4.3%)	578 (95.7%)
Controls	715	n.o.	15 (2.1%)	700 (97.9%).	15 (1.1%)	1415 (98.9%)
		(CC) vs. (GG) OR 95% CI *p*-value	(GC) vs. (GG) OR 95% CI *p*-value	(CC) + (GC) vs. (GG) OR 95% CI *p*-value	(C) vs. (G) OR 95% CI *p*-value	
IBD vs. controls		OR = 3.49 CI = (0.14–85.92) *p* = 0.280	OR = 1.09 CI = (0.04–28.30) *p* = 0.551	OR = 3.34 CI = (0.13–82.02) *p* = 0.291	**OR = 3.34** **CI = (1.85–6.02)** ***p* < 0.001**	
UC adults vs. controls		n.a.	OR = 0.89 CI = (0.02–47.36) *p* = 1.000	OR = 3.37 CI = (0.07–170.19) *p* = 1.000	**OR = 3.94** **CI = (1.95–7.96)** ***p* < 0.001**	
UC children vs. controls		n.a.	OR = 10.33 CI = (0.14–734.03) *p* = 1.000	OR = 10.92 CI = (0.21–555.02) *p* = 1.000	OR = 0.73 CI = (0.09–5.58) *p* = 1.000	
UC adults vs. UC children		n.a.	OR = 0.08 CI = (0.00–6.06) *p* = 1.000	OR = 0.31 CI = (0.00–15.69) *p* = 1.000	OR = 5.38 CI = (0.71–40.88) *p* = 0.089	
UC women vs. UC men		n.a.	OR = 1.27 CI = (0.02–71.63) *p* = 1.000	OR = 1.13 CI = (0.02–57.51) *p* = 1.000	OR = 0.87 CI = (0.32–2.39) *p* = 0.795	
UC girls vs. UC boys		n.a.	OR = 3.00 CI = (0.02–473.06) *p* = 1.000	OR = 1.32 CI = (0.02–68.34) *p* = 1.000	OR = 0.43 CI = (0.01–10.85) *p* = 1.203	
UC whole group vs. controls		OR = 2.16 CI = (0.04–109.03) *p* = 1.000	OR = 0.83 CI = (0.02–44.73) *p* = 1.000	OR = 2.09 CI = (0.04–105.50) *p* = 1.000	**OR = 2.55** **CI = (1.27–5.09)** ***p* = 0.006**	
CD adults vs. controls		OR = 10.38 CI = (0.42–255.72) *p* = 0.063	OR = 3.21 CI = (0.12–85.20) *p* = 0.309	OR = 9.91 CI = (0.40–244.26) *p* = 0.069	**OR = 3.61** **CI = (1.77–7.36)** ***p* < 0.001**	
CD children vs. controls		n.a.	OR = 1.47 CI = (0.02–80.39) *p* = 1.000	OR = 8.37 CI = (0.16–424.45) *p* = 1.000	**OR = 5.89** **CI = (2.60–13.34)** ***p* < 0.001**	
CD adults vs. CD children		OR = 1.12 CI = (0.04–27.76) *p* = 0.542	OR = 2.17 CI = (0.08–58.76) *p* = 0.405	OR = 1.18 CI = (0.05–29.37) *p* = 0.530	OR = 0.61 CI = (0.27–1.38) *p* = 0.232	
CD women vs. CD men		OR = 2.71 CI = (0.11–67.52) *p* = 0.342	OR = 1.28 CI = (0.04–37.98) *p* = 0.532	OR = 2.58 CI = (0.10–64.24) *p* = 0.354	OR = 2.65 CI = (0.85–8.35) *p* = 0.085	
CD girls vs. CD boys		n.a.	OR = 2.14 CI = (0.03–131.93) *p* = 1.000	OR = 1.07 CI = (0.02–55.28) *p* = 1.000	OR = 0.44 CI = (0.11–1.76) *p* = 0.332	
CD whole group vs. controls		OR = 7.57 CI = (0.31–186.46) *p* = 0.112	OR = 1.89 CI = (0.07–49.60) *p* = 0.433	OR = 7.11 CI = (0.29–175.26) *p* = 0.124	**OR = 4.24** **CI = (2.23–8.07)** ***p* < 0.001**	
CD whole group vs. UC whole group		OR = 3.50 CI = (0.14–86.46) *p* = 0.279	OR = 2.26 CI = (0.08–58.83) *p* = 0.390	OR = 3.40 CI = (0.14–83.97) *p* = 0.287	OR = 1.66 CI = (0.90–3.06) *p* = 0.099	

Statistically significant results are marked in bold; n.o.: not observed; n.a.: not analyzed (CC genotype was not detected in both juxtaposed groups); *: deviation from HWE.

**Table 4 jcm-10-03777-t004:** Allele and genotype distribution for *NOD2* c.2798+158C>T (rs5743289) polymorphism.

c.2798 + 158C>T rs5743289		Genotypes *n* (%)			Alleles *n* (%)	
Group	*n*	TT	TC	CC	T	C
IBD whole group	348	26 (7.5%)	108 (31.0%)	214 (61.5%)	160 (23.0%)	536 (77.0%)
UC adults	111	4 (3.6%)	24 (21.6%)	83 (74.8%)	32 (14.4%)	190 (85.6%)
UC adult women	57	2 (3.5%)	12 (21.1%)	43 (75.4%)	16 (14.0%)	98 (86.0%)
UC adult men	54	2 (3.7%)	12 (22.2%)	40 (74.1%)	16 (14.8%)	92 (85.2%)
UC children	62	n.o.	21 (33.9%)	41 (66.1%)	21 (16.9%)	103 (83.1%)
UC girls	25	n.o.	10 (40.0%)	15 (60.0%)	10 (20%)	40 (80%)
UC boys	37	n.o.	11 (29.7%)	26 (70.3%)	11 (14.9%)	63 (85.1%)
UC whole group	173	4 (2.3%)	45 (26.0%)	124 (71.7%)	53 (15.3%)	293 (84.7%)
CD adults	97	6 (6.2%)	38 (39.2%)	53 (54.6%)	50 (25.8%)	144 (74.2%)
CD adult women	49	4 (8.2%)	23 (46.9%)	22 (44.9%)	31 (31.6%)	67 (68.4%)
CD adult men	48	2 (4.2%)	15 (31.2%)	31 (64.6%)	19 (19.8%)	77 (80.2%)
CD children	78	16 (20.5%)	25 (32.1%)	37 (47.4%)	57 (36.5%)	99 (63.5%)
CD girls	39	7 (18.0%)	13 (33.3%)	19 (48.7%)	27 (34.6%)	51 (65.4%)
CD boys	39	9 (23.1%)	12 (30.8%)	18 (46.1%)	30 (38.5%)	48 (61.5%)
CD whole group	175	22 (12.6%)	63 (36.0%)	90 (51.4%)	107 (30.6%)	243 (69.4%)
Controls	231	10 (4.3%)	59 (25.5%)	162 (70.1%)	79 (17.1%)	383 (82.9%)
		(TT) vs. (CC) OR 95% CI *p*-value	(TC) vs. (CC) OR 95% CI *p*-value	(TT) + (TC) vs. (CC) OR 95% CI *p*-value	(T) vs. (C) OR 95% CI *p*-value	
IBD whole group vs. controls		OR = 1.97 CI = (0.92–4.20) *p* = 0.075	OR = 1.42 CI = (0.64–3.15) *p* = 0.386	OR = 1.47 CI = (1.03–2.10) *p* = 0.033	**OR = 1.45** **CI = (1.07–1.95)** ***p* = 0.015**	
UC adults vs. controls		OR = 0.78 CI = (0.24–2.58) *p* = 0.683	OR = 0.98 CI = (0.28–3.44) *p* = 0.979	OR = 0.79 CI = (0.47–1.32) *p* = 0.372	OR = 0.82 CI = (0.52–1.28) *p* = 0.373	
UC children vs. controls		OR = 0.19 CI = (0.01–3.25) *p* = 0.114	OR = 0.13 CI = (0.01–2.35) *p* = 0.064	OR = 1.20 CI = (0.66–2.18) *p* = 0.544	OR = 0.99 CI = (0.58–1.68) *p* = 0.966	
UC adults vs. UC children		OR = 4.47 CI = (0.24–85.07) *p* = 0.163	OR = 7.90 CI = (0.40–155.28) *p* = 0.071	OR = 0.66 CI = (0.33–1.30) P = 0.226	OR = 0.83 CI = (0.45–1.51) *p* = 0.532	
UC women vs. UC men		OR = 0.93 CI = (0.13-6.92) *p* = 0.944	OR = 1.00 CI = (0.12–8.31) *p* = 1.00	OR = 0.93 CI = (0.40–2.19) *p* = 0.869	OR = 0.94 CI = (0.44-1.99) *p* = 0.869	
UC girls vs. UC boys		OR = 1.71 CI = (0.03–90.56) *p* = 1.00	OR = 1.10 CI = (0.02–60.29) *p* = 1.00	OR = 1.58 CI = (0.54–4.58) *p* = 0.402	OR = 1.43 CI = (0.56–3.68) *p* = 0.455	
UC whole group vs. controls		OR = 0.52 CI = (0.16–1.71) *p* = 0.275	OR = 0.52 CI = (0.15–1.78) *p* = 0.295	OR = 0.93 CI = (0.66–1.43) *p* = 0.735	OR = 0.88 CI = (0.60–1.28) *p* = 0.498	
CD adults vs. controls		OR = 1.83 CI = (0.64–5.29) *p* = 0.256	OR = 0.93 CI = (0.31–2.77) *p* = 0.899	OR = 1.95 CI = (1.20–3.18) *p* = 0.007	**OR = 1.68** **CI = (1.13–2.52)** ***p* = 0.011**	
CD children vs. controls		**OR = 7.01** **CI = (2.94–16.67)** ***p* < 0.001**	**OR = 3.78** **CI = (1.51–9.46)** ***p* = 0.003**	**OR = 2.60** **CI = (1.54–4.40)** ***p* < 0.001**	**OR = 2.79** **CI = (1.86–4.19)** ***p* < 0.001**	
CD adults vs. CD children		**OR = 0.26** **CI = (0.09–0.73)** ***p* = 0.008**	**OR = 0.25** **CI = (0.09–0.72)** ***p* = 0.008**	OR = 0.75 CI = (0.41–1.36) *p* = 0.343	**OR = 0.60** **CI = (0.38–0.95)** ***p* = 0.030**	
CD women vs. CD men		OR = 2.82 CI = (0.47–16.76) *p* = 0.239	OR = 1.30 CI = (0.21–8.03) *p* = 0.774	OR = 2.24 CI = (0.99–5.06) *p* = 0.052	OR = 1.88 CI = (0.97–3.62) *p* = 0.059	
CD girls vs. CD boys		OR = 0.74 CI = (0.23–2.40) *p* = 0.611	OR = 0.72 CI = (0.20–2.54) *p* = 0.606	OR = 0.90 CI = (0.37–2.20) *p* = 0.821	OR = 0.85 CI = (0.44–1.63) *p* = 0.618	
CD whole group vs. controls		**OR = 3.96** **CI = (1.80–8.73)** ***p* < 0.001**	OR = 2.06 CI = (0.90–4.71) *p* = 0.083	**OR = 2.22** **CI = (1.47–3.34)** ***p* < 0.001**	**OR = 2.14** **CI = (1.53–2.98)** ***p* < 0.001**	
CD whole group vs. UC whole group		**OR = 7.58** **CI = (2.52–22.75)** ***p* < 0.001**	**OR = 3.93** **CI = (1.27–12.19)** ***p* = 0.013**	**OR = 2.39** **CI = (1.53–3.73)** ***p* < 0.001**	**OR = 2.43** **CI = (1.68–3.53)** ***p* < 0.001**	

Statistically significant values are marked in bold; n.o.: not observed.

**Table 5 jcm-10-03777-t005:** Allele and genotype distribution for the *NOD2* c.3020insC (p.1007fs rs5743293) variant.

c.3020insC (p.1007fs) rs5743293		Genotypes *n* (%)			Alleles *n* (%)	
Group	*n*	insC/insC	-/insC	-/-	-	insC
IBD whole group *	573	39 (6.8%)	90 (15.7%)	444 (77.5%)	978 (85.3%)	168 (14.7%)
UC adults *	210	6 (2.9%)	20 (9.5%)	184 (87.6%)	388 (92.4%)	32 (7.6%)
UC adult women *	98	4 (4.1%)	4 (4.1%)	90 (91.8%)	184 (93.9%)	12 (6.1%)
UC adult men	112	2 (1.8%)	16 (14.3%)	94 (83.9%)	204 (91.1%)	20 (8.9%)
UC children	64	1 (1.6%)	11 (17.2%)	52 (81.2%)	115 (89.8%)	13 (10.2%)
UC girls	27	1 (3.7%)	4 (14.8%)	22 (81.5%)	48 (88.9%)	6 (11.1%)
UC boys	37	n.o.	7 (19.9%)	30 (81.1%)	67 (90.5%)	7 (9.5%)
UC whole group *	274	7 (2.6%)	31 (11.3%)	236 (86.1%)	503 (91.8%)	45 (8.2%)
CD adults *	215	25 (11.6%)	48 (22.3%)	142 (66.1%)	332 (77.2%)	98 (22.8%)
CD adult women *	117	11 (9.4%)	21 (17.9%)	85 (72.7%)	191 (81.6%)	43 (18.4%)
CD adult men *	98	14 (14.3%)	27 (27.5%)	57 (58.2%)	141 (71.9%)	55 (28.1%)
CD children *	84	7 (8.3%)	11 (13.1%)	66 (78.6%)	143 (85.1%)	25 (14.9%)
CD girls *	40	3 (7.5%)	4 (10%)	33 (82.5%)	70 (87.5%)	10 (12.5%)
CD boys *	44	4 (9.1%)	7 (15.9%)	33 (75%)	73 (83%)	15 (17%)
CD whole group *	299	32 (10.7%)	59 (19.7%)	208 (69.6%)	475 (79.4%)	123 (20.6%)
Controls	888	n.o.	73 (8.2%)	815 (91.8%)	73 (4.1%)	1703 (95.9%)
		(insC/insC) vs. (-/-) OR 95% CI *p*-value	(insC/-) vs. (-/-) OR 95% CI *p*-value	(insC/insC) + (insC/-) vs. (-/-) OR 95% CI *p*-value	(insC) vs. (-) OR 95% CI *p*-value	
IBD vs. controls		**OR = 64.16** **CI = (8.38–2228.16)** ***p* < 0.001**	**OR = 2.26** **CI = (1.63–3.15)** ***p* < 0.001**	**OR = 3.24** **CI = (2.38–4.42)** ***p* < 0.001**	**OR = 3.01** **CI = (3.01–5.33)** ***p* < 0.001**	
UC adults vs. controls		**OR = 57.46** **CI = (3.22–1024.52)** ***p* < 0.001**	OR = 1.21 CI = (0.72–2.04) *p* = 0.465	OR = 1.58 CI = (0.98–2.54) *p* = 0.058	**OR = 1.92** **CI = (1.25–2.96)** ***p* = 0.002**	
UC children vs. controls		**OR = 46.60** **CI = (1.87–1157.89)** ***p* < 0.001**	**OR = 2.36** **CI = (1.18–4.72)** ***p* = 0.013**	**OR = 2.58** **CI = (1.32–5.04)** ***p* = 0.004**	**OR = 2.64** **CI = (1.42–4.90)** ***p* = 0.001**	
UC adults vs. UC children		OR = 1.70 CI = (0.20–14.40) *p* = 0.625	OR = 0.51 CI = (0.23–1.41) *p* = 0.097	OR = 0.61 CI = (0.29–1.30) *p* = 0.197	OR = 0.73 CI = (0.37–1.44) *p* = 0.360	
UC women vs. UC men		OR = 2.10 CI = (0.37–11.69) *p* = 0.392	**OR = 0.26** **CI = (0.08–0.81)** ***p* = 0.014**	OR = 0.46 CI = (0.19–1.12) *p* = 0.083	OR = 0.67 CI = (0.32–1.40) *p* = 0.280	
UC girls vs. UC boys		OR = 4.07 CI = (0.16–104.53) *p* = 0.249	OR = 0.78 CI = (0.20–2.99) *p* = 0.716	OR = 0.97 CI = (2.84–3.48) *p* = 0.968	OR = 1.20 CI = (0.38–3.79) *p* = 0.760	
UC whole group vs. controls		**OR = 51.72** **CI = (2.94–908.93)** ***p* < 0.001**	OR = 1.47 CI = (0.94–3.29) *p* = 0.090	**OR = 1.80** **CI = (0.18–2.73)** ***p* = 0.005**	**OR = 2.09** **CI = (1.42–3.07)** ***p* < 0.001**	
CD adults vs. controls		**OR = 291.86** **CI = (17.67–4821.24)** ***p* < 0.001**	**OR = 3.77** **CI = (2.52–5.66)** ***p* < 0.001**	**OR = 5.74** **CI = (3.96–8.31)** ***p* < 0.001**	**OR = 6.89** **CI = (4.98–9.53)** ***p* < 0.001**	
CD children vs. controls		**OR = 183.95** **CI = (10.39–3255.99)** ***p* < 0.001**	OR = 1.86 CI = (0.94–3.70) *p* = 0.070	**OR = 3.05** **CI = (1.72–5.40)** ***p* < 0.001**	**OR = 4.08** **CI = (2.51–6.63)** ***p* < 0.001**	
CD adults vs. CD children		OR = 1.66 CI = (0.68–4.03) *p* = 0.259	OR = 2.03 CI = (0.99–4.16) *p* = 0.050	**OR = 1.86** **CI = (1.04–3.41)** ***p* = 0.034**	**OR = 1.69** **CI = (1.04–2.73)** ***p* = 0.032**	
CD women vs. CD men		OR = 0.53 CI = (0.22–1.24) *p* = 0.139	OR = 0.52 CI = (0.27–1.01) *p* = 0.052	**OR = 0.52** **CI = (0.30–0.93)** ***p* = 0.026**	**OR = 0.58** **CI = (0.37–0.91)** ***p* = 0.017**	
CD girls vs. CD boys		OR = 0.75 CI = (0.16–3.62) *p* = 0.719	OR = 0.57 CI = (0.15–2.14) *p* = 0.402	OR = 0.64 CI = (0.22–1.84) *p* = 0.402	OR = 0.70 CI = (0.29–1.65) *p* = 0.408	
CD whole group vs. controls		**OR = 254.23** **CI = (15.50–4169.02)** ***p* < 0.001**	**OR = 3.17** **CI = (2.18–4.61)** ***p* < 0.001**	**OR = 4.88** **CI = (3.46–6.89)** ***p* < 0.001**	**OR = 6.04** **CI = (4.44–8.21)** ***p* < 0.001**	
CD whole group vs. UC whole group		**OR = 5.19** **CI = (2.24–12.00)** ***p* < 0.001**	**OR = 2.16** **CI = (1.35–3.47)** ***p* = 0.001**	**OR = 2.72** **CI = (1.78–4.14)** ***p* < 0.001**	**OR = 2.89** **CI = (2.01–4.16)** ***p* < 0.001**	

Statistically significant results are marked in bold; n.o.: not observed; *: deviation from HWE.

**Table 6 jcm-10-03777-t006:** *NOD2* gene haplotypes and association analysis.

Name *	Alleles/Haplotypes					Frequency **			Disease Entity CD vs. UC		Crohn’s Disease						Ulcerative Colitis					
											Predisposition to Late-Onset Form		Predisposition to Early-Onset Form		Sex (Women vs. Men)		Predisposition to Late-Onset Form		Predisposition to Early-Onset Form		Sex (Women vs. Men)	
	p.P268S	p.R702W	p.G908R	c.2798 +158C>T	1007fs	Controls	CD	UC	OR (95% CI)	*p*-Value	OR (95% CI)	*p*-Value	OR (95% CI)	*p*-Value	OR (95% CI)	*p*-Value	OR (95% CI)	*p*-Value	OR (95% CI)	*p*-Value	OR (95% CI)	*p*-Value
A	C	C	G	C	wt.	39.5%	50.3%	72.0%	**0.40** **(0.32–0.51)**	**< 0.001**	**1.77** **(1.27–2.48)**	**< 0.001**	1.26 (0.87–1.82)	0.241	0.61 (0.35–1.08)	0.091	**3.40** **(2.43–4.78)**	**< 0.001**	**2.67** **(1.77–4.01)**	**< 0.001**	1.17 (0.66–2.07)	0.598
-	C	**C**	C	C	wt.	25.4%	n.o.	n.o.	n.a.	-	**0.01** **(0.00–0.12)**	**< 0.001**	**0.03** **(0.01–0.16)**	**< 0.001**	n.a.	-	**0.03** **(0.01–0.11)**	**< 0.001**	**0.02** **(0.01–0.17)**	**< 0.001**	n.a.	-
H	**T**	C	G	C	wt.	8.2%	9.7%	7.2%	1.24 (0.81–1.88	0.323	1.35 (0.77–2.35)	0.297	1.44 (0.80–2.59)	0.195	0.75 (0.29–1.96)	0.561	1.04 (0.59–1.85)	0.883	1.74 (0.95–3.19)	0.074	0.67 (0.25–1.84)	0.440
D	**T**	C	**C**	C	wt.	5.2%	3.1%	1.7%	1.88 (0.90–3.93)	0.095	**0.10** **(0.01–0.72)**	**0.023**	0.50 (0.17–1.46)	0.206	7.69 (0.39–150.88)	0.179	**0.34** **(0.12–1.00)**	**0.051**	0.15 (0.02–1.11)	0.063	1.49 (0.24–9.07)	0.668
–	**T**	C	**C**	**T**	wt.	4.3%	n.o.	n.o.	n.a.	-	**0.12** **(0.02–0.88)**	**0.037**	0.07 (0.01–1.19)	0.066	n.a.	–	**0.10** **(0.01–0.77)**	**0.027**	0.09 (0.01–1.46)	0.090	n.a.	-
G	**T**	C	G	**T**	wt.	4.2%	9.0%	6.8%	1.28 (0.82–1.99)	0.283	1.89 (0.95–3.78)	0.071	**2.97** **(1.50–5.88)**	**0.002**	3.18 (0.98–10.37)	0.055	**2.00** **(1.04–3.87)**	**0.039**	**2.61** **(1.26–5.41)**	**0.010**	1.51 (0.56–4.05)	0.412
E	**T**	C	G	**T**	**insC**	3.5%	13.3%	3.3%	**4.13** **(2.55–6.71)**	**< 0.001**	**5.09** **(2.59–9.97)**	**< 0.001**	**4.62** **(2.26–9.47)**	**< 0.001**	1.28 (0.56–2.94)	0.555	0.76 (0.27–2.13)	0.598	1.92 (0.76–4.85)	0.170	0.65 (0.11–3.96)	0.639
-	**T**	C	**C**	**T**	**insC**	2.1%	n.o.	n.o.	n.a.	-	0.11 (0.01–1.95)	0.134	0.61 (0.13–2.82)	0.527	n.a.	-	0.10 (0.01–1.70)	0.111	0.10 (0.01–3.00)	0.229	n.a.	-
-	C	C	G	C	**insC**	1.9%	1.6%	2.0%	0.70 (0.28–1.75)	0.447	1.08 (0.33–3.56)	0.897	1.02 (0.27–3.82)	0.977	0.53 (0.05–5.91)	0.603	2.17 (0.85–5.55)	0.105	0.41 (0.05–3.29)	0.404	1.24 (0.32–4.74)	0.755
-	C	C	**C**	C	**insC**	1.7%	n.o.	n.o.	n.a.	-	0.14 (0.01–2.46)	0.179	0.18 (0.01–3.10)	0.274	n.a.	-	0.12 (0.01–2.15)	0.151	0.22 (0.01–3.78)	0.295	n.a.	-
-	C	C	**C**	**T**	wt.	1.4%	n.o.	n.o.	n.a.	-	0.16 (0.01–2.82)	0.211	0.20 (0.01–3.55)	0.274	n.a.	-	0.14 (0.01-2.46)	0.179	0.25 (0.01–4.34)	0.339	n.a.	-
-	**T**	C	G	C	**insC**	n.o.	2.8%	1.9%	**2.47** **(1.08–5.64)**	**0.033**	**23.01** **(2.90–182.91)**	**0.003**	n.a.	-	1.08 (0.30–3.85)	0.907	n.a.	-	**3.73** **(1.06–13.08)**	**0.040**	n.a.	-
B	**T**	**T**	G	**T**	wt.	n.o.	5.1%	1.6%	**2.90** **(1.40–6.00)**	**0.004**	**17.71** **(2.16–144.90)**	**0.007**	**29.36** **(3.69–233.68)**	**0.001**	3.34 (0.66–16.98)	0.146	n.a.	-	1.81 (045–7.35)	0.405	n.a.	-
-	C	C	G	**T**	wt.	n.o.	2.4%	n.o.	1.33 (0.53–3.34	0.537	n.a.	-	**12.93** **(2.72–61.58)**	**0.001**	n.a.	-	n.a.	-	n.a.	-	n.a.	-

Statistically significant results and minor alleles are marked in bold; wt.: wild type allele (in this case: allele without insertion); *: according to Tukel et al.; **: haplotype frequencies evaluated and rounded using Haploview; n.a.: not analyzed; n.o.: not observed.

**Table 7 jcm-10-03777-t007:** Overview of the frequency of *NOD2* gene variants in different ethnic groups compared to the Polish population (analyzed in this study).

	p.P268S (rs2066842)	p.R702W (rs2066844)	p.G908R (rs2066845)	IVS8 (rs5743289)	3020insC (rs2066847)
Global (gnomAD)	18.6%	2.6%	0.0%	10.2%	1.7%
Polish (experimental data)	30.1%	2.0%	1.1%	17.1%	4.1%
Ashkenasi Jews (gnomAD)	23.9%	2.2%	0.4%	16.9%	3.4%
Estonian (dbSNP)	20.7%	1.9%	0.7%	12.1%	3.8%
Northern Sweden (dbSNP)	24.0%	1.2%	0.6%	17.0%	1.7%
East Asian (gnomAD)	0.5%	0.0%	0.0%	0.0%	0.0%
African (gnomAD)	5.0%	0.7%	0.0%	2.9%	0.3%
American (gnomAD)	16.0%	0.3%	1.0%	10.0%	1.0%

## Data Availability

The data presented in this study are available on request from the corresponding author.

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
