# Peer review of "Crohn’s Disease Susceptibility and Onset Are Strongly Related to Three NOD2 Gene Haplotypes"

_jcm, 2021, doi:10.3390/jcm10173777_

Round 1

Reviewer 1 Report

The manuscript is well written and rich in technical details and appropriate references. Reported data are accurately explained with their statistical value. Maybe the tables, complex and full of numbers, could be revised for a simplier reading, possibly with the help of a graph.

The epidemiological considerations are interesting and suggest furterello complex correlation with the clinical outcome, including need for surgery, interval time between diseases riacutizations, drug responding profile and so on.

Author Response

We would like to thank the Reviewer for feedback and comments.

The tables are voluminous and may appear challenging to analyze. However, we have not been able to simplify them without losing statistically necessary data for presenting this type of result. We considered presenting a graph but the volume of information required to present it, such as confidence intervals for odds ratios, meant it was not as readable as expected. We prepared a graph that reflected differences between studied groups (presented in attachment). However, we decided not to include it in the article's final version as the frequencies were presented in table 7 accompanied by odds ratios, which is more comprehensive, and we were afraid to double the presented data.

Reviewer 2 Report

even if the contribution of NOD2 in CD in known the manuscript is interesting and well presented. 

It will be interesting to stratify the genotypes also with disease subphenotypes

and to include and discuss related papers ie World J Gastroenterol. 2010 Apr 14;16(14):1753-8. World J Gastroenterol. 2005 Feb 7;11(5):681-5. Eur J Gastroenterol Hepatol. 2004 Nov;16(11):1177-82.

Author Response

We appreciate the favorable revision very much and tried to improve the presented article.

We agree that stratifying genotypes and haplotypes would be very interesting, but we did not dispose of complete and detailed clinical data. Our institute is not a medical unit, and we do not have access to medical databases. We cooperate with physicians from several hospitals in Poland, but the amount of data we obtain is usually limited. We fight to gather information, but it is not easy, especially in the data protection policy era.

We are thankful for indicating papers reporting NOD2 mutations in the Greek population that we somehow omitted in the literature search, and we inserted fragment in the discussion section with appropriate references in lines 377-380:

"In Greek CD patients, the 1007finsC mutation was significantly more frequent in childhood-onset than in adult-onset form [36]. Other research carried out in Greece pointed out the association of R702W, G908R, and 1007insC with ileitis or ileocolitis in CD clinical picture [37,38]."